# Services, models of care, and interventions to improve access to cancer treatment for adults who are socially disadvantaged: A scoping review protocol

Tara C. Horrill[1]*, Amber Bourgeois[2], Max Kleijberg[3], Janice Linton[4], Kate Leahy[2], Kelli I. Stajduhar[2]

1 College of Nursing, University of Manitoba, Winnipeg, Manitoba, Canada, 2 School of Nursing, University of Victoria, Victoria, British Columbia, Canada, 3 Karolinska Institutet, Fleminsberg, Sweden, 4 Neil John Maclean Health Sciences Library, University of Manitoba, Winnipeg, Manitoba, Canada

* tara.horrill@umanitoba.ca

**Data Availability Statement:** N/A.

## Abstract

Timely access to guideline-recommended cancer treatment is known to be an indicator of the quality and accessibility of a cancer care system. Yet people who are socially disadvantaged experience inequities in access to cancer treatment that have significant impacts on cancer outcomes and quality of life. Among people experiencing the intersecting impacts of poor access to the social determinants of health and personal identities typically marginalized from society ('social disadvantage'), there are significant barriers to accessing cancer, many of which compound one another, making cancer treatment extremely difficult to access. Although some research has focused on barriers to accessing cancer treatment among people who are socially disadvantaged, it is not entirely clear what, if anything, is being done to mitigate these barriers and improve access to care. Increasingly, there is a need to design cancer treatment services and models of care that are flexible, tailored to meet the needs of patients, and innovative in reaching out to socially disadvantaged groups. In this paper, we report the protocol for a planned scoping review which aims to answer the following question: What services, models of care, or interventions have been developed to improve access to or receipt of cancer treatment for adults who are socially disadvantaged? Based on the methodological framework of Arksey and O'Malley, this scoping review is planned in six iterative stages. A comprehensive search strategy will be developed by an academic librarian. OVID Medline, EMBASE, CINAHL (using EBSCOhost) and Scopus will be searched for peer-reviewed published literature; advanced searches in Google will be done to identify relevant online grey literature reports. Descriptive and thematic analysis methods will be used to analyze extracted data. Findings will provide a better understanding of the range and nature of strategies developed to mitigate barriers to accessing cancer treatment.

**Funding:** This work was supported by a grant from the Manitoba Center for Nursing and Health Research at the University of Manitoba (UM #58850; Horrill: Principle Applicant; all other authors: Co-Applicants). The funders had no role in the study design, data collection and analysis, decision to publish, or preparation of the manuscript.

**Competing interests:** The authors have no competing interests to declare.

## Introduction

Timely access to guideline-recommended cancer treatment is known to be an indicator of the quality and accessibility of a cancer care system [1]. Yet people who are socially disadvantaged experience inequities in access to cancer treatment that have significant impacts on cancer outcomes and quality of life. We understand social disadvantage as an outcome of the complex relationships between access to the social determinants of health (e.g., income, education, housing) and aspects of identity (e.g., 'race', gender, disability) that structure peoples' social positioning. People who are socially disadvantaged are often pushed to the margins and/or excluded from society in multiple ways, resulting in less access to resources (including healthcare) and as a result, fewer opportunities for health [2]. Broader contexts also intersect to impact social disadvantage (e.g., public policies); the complex relationships between social determinants of health, social positioning, and broader social, economic, political, and historical contexts are explicated in the WHO's Commission on the Social Determinants of Health [3], and the Conceptual Framework for Action on the Social Determinants of Health that followed [4].

In the context of social disadvantage and cancer treatment, adults experiencing the highest levels of material deprivation (a proxy measure for social disadvantage incorporating income, employment, and education) are significantly less likely to be seen by a medical or radiation oncologist after a cancer diagnosis [5]. This is particularly problematic, given that consultations with medical or radiation oncology often represent the entry point into the cancer care system [5]. As a result of their social positioning, people who are socially disadvantaged by way of unstable housing or homelessness often experience significant delays in initiating cancer treatment [6, 7], are less likely to receive treatment for their cancer [7–9] or are offered treatment that is of lower quality (i.e., does not align with current standard of care) [10]. Evidence also suggests that race and/or ethnicity are closely linked to social disadvantage and are associated with inequities in cancer treatment. For example, a recent review of cancer inequities in the United States found that non-white men were more likely to: encounter delays in treatment for prostate cancer, receive poorer quality treatment, and experience more side effects during and after treatment, with even worse outcomes for non-white men with low socioeconomic status, suggesting a compounding effect [11]. Inequities in cancer treatment are reflected in cancer outcomes, and the wealth of evidence demonstrating persistent disparities in cancer mortality, survival, and quality of life among socially disadvantaged people, both within and between countries [12–14].

Underlying these inequities are significant barriers to accessing cancer treatment among people who are socially disadvantaged. Lack of access to safe, stable, and affordable housing complicates the receipt of cancer treatment, and care coordination during treatment [15]. For people who are unstably housed, meeting daily needs for shelter and food take priority, and may delay cancer treatment [16, 17]. Social and economic disadvantage are often linked to lower levels of health literacy, which presents specific challenges to cancer treatment, including patients who do not understand their treatment, resulting in skipped or missed treatments, delays in treatment, or decisions to decline treatment [16]. The design and delivery of many cancer treatment services also creates barriers: inflexible services and standardized, protocol-driven care mean that people who are socially disadvantaged, particularly those living in extreme poverty and experiencing homelessness, are not able to initiate cancer treatment or are not able to complete treatment that has been initiated [18]. Evidence from our ethnographic research in progress suggests social disadvantage often stems from and intersects with experiences of racism, discrimination and stigma related to substance use (actual or perceived) and mental health, resulting in patients who have actual or anticipated negative experiences in

the cancer system, and avoid or delay accessing care. Moreover, for some patients, attending appointments to access cancer treatment within an institutionalized cancer center (e.g., hospital) can evoke a trauma response, further complicating their cancer treatment experience. Many of these barriers intersect and compound one another, making cancer treatment extremely difficult to access for people who are socially disadvantaged [16, 19].

## Objectives

Although some research has focused on barriers to accessing cancer treatment among people who are socially disadvantaged, it is not entirely clear what, if anything, is being done to mitigate these barriers and improve access to care. In our programs of research, we are increasingly seeing the necessity of designing cancer treatment services and models of care that are flexible, tailored to meet the needs of patients, and innovative in reaching out to socially disadvantaged groups. In an ongoing ethnographic study, interviews with outreach, health and social service providers who work closely with socially disadvantaged adults have repeatedly identified such services as urgently needed for their patients diagnosed with cancer. Although our programs of research are primarily located within the Western context (North America and Europe), we are interested in learning from and about services and models of care within geographically diverse contexts. The purpose of this scoping review is to explore cancer treatment services and models of care designed to improve access to or receipt of cancer treatment among people who are socially disadvantaged. Preliminary literature searches conducted by the academic librarian (JL) indicate there are no other planned or ongoing reviews on this topic.

## Theoretical perspectives/framework

Our review is grounded in critical theoretical and social justice perspectives. Through these perspectives, health is understood as a basic human right, and as influenced by multiple contexts, including sociocultural, economic, political and historical contexts [20, 21]. From a critical theoretical perspective, health inequities exemplify social *in*justice, and are understood as differences in health that are "socially produced, systematic in their distribution across a population, and unfair" [4 p.12]. Health inequities are both created and maintained by social and structural determinants of health, including material circumstances and living conditions, socioeconomic position, education, income, social and public policies. The health system, and access to healthcare are understood as important intermediary determinants of health [4], and are of particular interest to health researchers, and leaders and clinicians working within the health system.

An intersectional theoretical perspective also informs this review. From an intersectional perspective, various aspects of identity and social location (e.g., 'race', gender, age, disability) and forms of oppression (e.g., systemic racism, colonialism) intersect and compound one another to situate people with varying levels of advantage or disadvantage in complex and nuanced ways [22]. Intersectional perspectives center social justice concerns, and facilitate exploration of the underlying causes of health and social inequities. In this review, are particularly interested in people who are disadvantaged in multiple intersecting ways, which often create a web of barriers complicating access to cancer treatment.

## Materials and methods

Our scoping review approach is based on the framework proposed by Arksey and O'Malley [23] and enhancements by Levac et al. [24]. As such, our review is planned in six iterative stages, as outlined below. This review protocol is reported based on the guidelines developed by Moher et al. [25], as outlined in the Preferred Reporting Items for Systematic Review and

Meta-Analysis Protocols (PRISMA-P). This protocol has been registered in Open Science Framework (https://doi.org/10.17605/OSF.IO/83ANZ). Any amendments to our protocol will be detailed in our report of review findings.

## Identifying the research question

Given the multitude of barriers to accessing cancer treatment described above, and the demonstrated need for cancer treatment services that are low-barrier, flexible, and designed to meet the needs of people who are socially disadvantaged, this scoping review aims to answer the following research question: "What services, models of care, or interventions have been developed to improve access to or receipt of cancer treatment for adults who are socially disadvantaged?" Our objectives are two-fold:

1. To identify the extent and summarize the nature of services, models of care, or interventions that have been developed and implemented with the explicit aim of improving access to cancer treatment for adults who are socially disadvantaged, including how they are designed, organized and delivered, and the types of strategies used to improve access to cancer treatment;

2. To summarize whether and how these services or models of care have been evaluated, and key findings from evaluations.

Key concepts within our research question include 'cancer treatment' and 'socially disadvantaged populations'. For the purposes of this review, cancer treatment is defined as treatments used to stop, shrink, or slow down the progression of cancer, including surgery, radiotherapy, systemic therapy (including chemotherapy, targeted therapy, immunotherapy and hormone therapy), stem cell and bone marrow transplant [26, 27]. We define social disadvantage as one's relative position in the social order, influenced and stratified by access to the social determinants of health (e.g., income, employment, housing) as well as aspects of one's identity, including skin color, national origin, religion, age, gender, gender identity, sexual orientation, disability, or mental health status [2, 28]. In this review, we are interested in people who are marginalized by way of one or more forms of social disadvantage, recognizing that how social disadvantage manifests may differ depending on context, but crosses geographical borders, and is not limited to a particular nation or region. Given that children and youth who are socially disadvantaged have entirely different experiences of disadvantage than adults, and that cancer care organizations structure adult and pediatric cancer services very differently, our focus in this review is on adults.

## Identifying relevant studies

A comprehensive search strategy has been designed by an academic librarian (JL). Primary databases to be searched include OVID Medline, EMBASE, and CINAHL (using EBSCOhost). We will search for additional articles using Scopus. Dissertation databases will be searched. Advanced searches in Google will be done to identify relevant online grey literature reports. Backwards and forwards searching of key full text articles will also be carried out to identify relevant publications not retrieved from the database searches. No date limits will be set but retrieval of grey literature and dissertations will be limited to those accessible online, so will have likely been published over the past twenty years. The review team (including the medical librarian) will search for full text across all available databases and where full text is unavailable, record/study will be excluded and reported as inaccessible. Search results will be merged and de-duplicated using Covidence software. A sample search strategy is included in S1 Appendix.

**Table 1. Inclusion and exclusion criteria.**

| Criteria | Inclusion | Exclusion |
|---|---|---|
| Language of Publication | English | All others |
| Population | Experiencing social disadvantage along one or more axis, including (but not limited to):<br>• Economic status (low income, poverty)<br>• Educational attainment<br>• Unstably housed<br>• Member of a racialized group<br>• Geography (rural, remote)<br>• Marginalized by way of: religion, national origin, immigrant status, gender, gender identity, sexual orientation, disability, mental health status, or substance use<br>Adult (18+) | Not experiencing social disadvantage<br><br>All others |
| Focus of Publication | • Focus is on describing a service, intervention or models of care developed and implemented with the explicit aim of *improving access* to or *receipt of* cancer treatment<br>• Cancer treatment: treatments to stop, shrink or slow the progression of cancer, including surgery, radiotherapy, systemic therapy (chemotherapy, targeted therapy, immunotherapy, and/or hormone therapy), stem cell transplant, or bone marrow transplant.<br>• The described service or model of care has been evaluated and evaluation data is available | |
| Type of publication | • Peer-reviewed primary research studies (including observational studies, randomized controlled trials, quasi-experimental studies and other non-randomized trials, systematic reviews, qualitative studies).<br>• Peer-reviewed quality improvement reports | Non-peer reviewed research, case reports, discussion papers, opinion papers, commentaries, theses/dissertations, conference proceedings, slide presentations, news stories |

## Study selection

We anticipate that study selection will be iterative, with inclusion and exclusion criteria evolving as we refine our search, and review and discuss articles for inclusion [24]. Initial inclusion and exclusion criteria are identified in Table 1. We will use a 3-step process for study selection. First, single reviewer will screen titles of all articles identified through electronic searches. Second, two reviewers will independently screen abstracts of remaining articles against the inclusion criteria. Third, two reviewers will independently screen full texts of articles included in step 2 for eligibility, noting the reason for exclusion, where applicable. Studies published in languages other than English will be excluded. This will unlikely have a major impact on the overall study findings as was found in a recent study, where the exclusion of non-English papers from reviews had minimal impact on overall review conclusions [29].

Any discrepancies between reviewers will be resolved by discussion, and if necessary, consultation with the PI. Study selection will be managed using Covidence software (www.covidence.org). Study selection will be reported as per the PRISMA extension for scoping reviews [30].

## Charting, summarizing, and reporting the results

Similar to study selection, we anticipate that the charting (extracting) of data will be iterative in nature [24], and will be managed using Covidence software. Two reviewers will independently extract data on the first five publications using the data charting form developed by the research team to pilot test the form. The team will then meet to compare data charted, and determine any necessary revisions to the form. After pilot testing the form, data charting will

continue, with regularly scheduled meetings to discuss issues arising, ensure data charting is consistent between reviewers, and meets the aims of the review. We have developed an initial data charting form (S2 Appendix) that includes data related to the publication, study design (if applicable), information on and detailed description of the service or model of care to improve access to cancer treatment, barriers addressed through or by the service or model of care, how it has been evaluated, and results of the evaluation.

To organize and summarize the data, we will first conduct a descriptive numerical analysis of study findings (e.g., types of publications, year of publication, type of target population). Given the primary objectives of the review, we will conduct a qualitative descriptive and/or thematic analysis of findings (e.g., types of services or models of care; how and what barriers they address). Findings will be reported as a narrative summary of findings, followed by a discussion of key considerations for health policy, health services delivery, and research. We anticipate that analysis will again be an iterative process: descriptive and thematic analysis will be primarily conducted by the PI, with assistance from a research assistant, with findings (themes) and implications discussed as a team, and refinements made.

Scoping review findings will be prepared for dissemination to both academic and non-academic audiences. Findings will be published in an academic, peer-reviewed, open access journal, and presented at relevant local, national and international conferences. To reach stakeholders and knowledge users, a non-technical report on the review findings, written for health service and policy leaders, will be developed. The non-technical report will be accompanied by an infographic, which will also be disseminated online via social media; both will be freely available online through our respective research program websites. In addition, our integrated knowledge translation approach (described below), will support knowledge transfer and exchange between the research team, stakeholders, and knowledge users at key junctures in the review.

Levac and colleagues [24] recommend a final consultation stage, in which preliminary findings are used as a foundation to inform consultations with stakeholders and knowledge users, who then have an opportunity to inform the findings and implications of the review. This review is being conducted within the context of multiple synergistic projects and programs of research focused on advancing health equity in cancer care. Through our programs of research, we regularly engage with stakeholders and knowledge users, both formally and informally, to inform our ongoing work. We will seek opportunities to discuss preliminary findings with stakeholders and knowledge users, including organizational and policy leaders, clinicians, and service users, with the aim of facilitating dialogue on our findings and implications for research, policy and practice. We envision dialogues to be an opportunity for reciprocal knowledge transfer and exchange, and as foundational to informing our future work.

## Limitations

The results of our review may be limited in several ways. First, as per scoping review guidelines, we do not intend to conduct quality appraisals of included publications, and although we do intend to describe whether and how services or models of care were evaluated, we do not intend to appraise the strength or quality of the evidence. Thus, insights about the services or models of care may be limited. Despite these potential limitations, our review will be strengthened by our strong interdisciplinary team with significant content and methodological expertise, and the nesting of this review within a program of research explicitly focused on advancing equity in cancer care, using multiple methods and community engagement, increasing the relevance of our review. The rigour of this review is enhanced by following an established methodology, a clearly described protocol, and reporting as per PRISMA-ScR

guidelines. Finally, ongoing relationships with stakeholders through multiple synergistic projects which will strengthen our findings, and their dissemination.

## Discussion

Emerging evidence has documented the scope and nature of barriers to accessing cancer treatment for people who are socially disadvantaged. This evidence points to the distinct need for innovative services or models of care that are purposefully designed to address specific barriers to accessing cancer treatment. As a first step towards redressing inequitable access to cancer treatment, there is a clear need to describe and map existing services or models of care which could be implemented elsewhere in the service of improving access to treatment.

## Conclusion

This review will result in a better understanding of the range and nature of strategies developed to mitigate barriers to accessing cancer treatment, and will lay a foundation for our team's future work focused on designing, implementing and evaluating strategies to purposefully address inequities in access to cancer treatment and care. Our team is engaged in multiple synergistic projects, in collaboration with socially disadvantaged people and the health and social service providers who work closely with them, as well as leaders and clinicians in the cancer care sector, and the results of this review will be informing our ongoing research projects as well as our efforts to determine research priorities.

## Supporting information

**S1 Appendix. Sample search strategy.**
(DOCX)

**S2 Appendix. Preliminary data charting form.**
(DOCX)

**S1 Checklist. PRISMA-P 2015 checklist.**
(DOCX)

## Author Contributions

**Conceptualization:** Tara C. Horrill, Amber Bourgeois, Max Kleijberg, Janice Linton, Kate Leahy, Kelli I. Stajduhar.

**Methodology:** Tara C. Horrill, Janice Linton.

**Project administration:** Tara C. Horrill.

**Supervision:** Tara C. Horrill.

**Writing – original draft:** Tara C. Horrill.

**Writing – review & editing:** Amber Bourgeois, Max Kleijberg, Janice Linton, Kate Leahy, Kelli I. Stajduhar.

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
