## [Decision Letter · Decision Letter 0]

29 Jun 2023

PONE-D-23-11579Services, models of care, and interventions to improve access to cancer treatment for people who are socially disadvantaged: A scoping review protocolPLOS ONE

Dear Dr. Horrill,

Thank you for submitting your manuscript to PLOS ONE. After careful consideration, we feel that it has merit but does not fully meet PLOS ONE’s publication criteria as it currently stands. Therefore, we invite you to submit a revised version of the manuscript that addresses the points raised during the review process.

We look forward to receiving your revised manuscript.

Kind regards,

Portia Janine Jordan, PhD

Academic Editor

PLOS ONE

Journal Requirements:

4. We note that you have referenced (ie. Horrill, T. C., et al. [15]) which has currently not yet been accepted for publication. Please remove this from your References and amend this to state in the body of your manuscript: (ie “Horrill, T. C., et al. [Unpublished]”) as detailed online in our guide for authors

**Additional Editor Comments:**

Dear Dr. Horrill,

We thank you for the submission of your manuscript. The peer review process of your manuscript has now been completed and we have reached a decision regarding your decision.

The manuscript is suitable for publication but needs minor adjustments.

The comments from the two reviewers are attached to the email.

Reviewer 1:

Thank you for allowing me to review this manuscript related to a protocol for a scoping review. The following may assist in improving the manuscript further:

Background: Although the concept of people who are socially disadvantaged is well-explained in the inclusion/exclusion criteria in the methodology, perhaps a definition early in the background could be provided as the concept of people who are socially disadvantaged could have different meanings in different contexts.

Objectives: the justification for conducting the scoping review is clear. However, this justification could be strengthened by indicating whether any reviews have been conducted on the topic or related topics and where gaps are.

Methods: although the inclusion and exclusion are well-described in terms of the concept and language, perhaps also to include what type of articles/studies/literature will be searched for.

Limitations: limitations were clearly indicated. Perhaps also to indicate what measures/strategies will be implemented to enhance rigour of the scoping review.

Documentation/reporting: the how of reporting of the review was outlined, however perhaps the reporting tools (e.g. PRISMA flowchart, tables, graphs) could be included.

Reviewer 2:

COMMENTS TO THE AUTHORS

This scoping review seeks to identify the services, models of care and interventions developed to improve access to cancer treatment for socially disadvantaged people. The study is relevant and can contribute to overcoming the barriers that socially disadvantaged people face.

However, the protocol requires minor modification/revision (specified below) before consideration for publication.

MAJOR ISSUES

Line 146: Indicate whether studies will be included or excluded based on their design and provide a rationale. For instance, will the authors include both experimental and non-experimental interventions that improve access to cancer treatment?

Line 155: Study selection: Describe how duplicate records/studies will be identified and managed.

Line 166: Table 1 (Inclusion and Exclusion Criteria) suggests that the review will focus on adults aged 18 years and older who are socially disadvantaged. This focus must reflect in the background, abstract and possibly the title of the review. Socially disadvantaged people may include children, adolescents or youth etc.

Line 166: Focus of publication: The authors indicated that “Focus is on describing a service, intervention or models of care developed and implemented with the explicit aim of improving access to, receipt of, or adherence to cancer treatment”. The concept “adherence” is introduced here for the first time in the protocol. Adherence to cancer treatment is absent in the review question, purpose or search strategy. Countless services, models of care and interventions (experimental/non-experimental) that “improve adherence to cancer treatment” exist and will considerably expand the scope and resources required of the review.

Consider deleting the “adherence to cancer treatment” component. Otherwise, comprehensively integrate the “adherence to cancer treatment” component into the scoping review.

MINOR ISSUES

Line 36: Review the sentence “A comprehensive search strategy will be developed by an academic librarian will be developed”.

Line 42: Consider removing the open science registration link from the abstract.

Line 55: Review sentence starting with “Evidence also….”

Line 146: Identifying relevant studies: Indicate what action(s) will be taken if the full text of a record/study (other than grey literature and dissertations) is not found.

Table 166: Exclusion criteria: Provide a rationale for studies published in languages other than English language.

Line 166: Consider merging the two population rows.

Reviewers' comments:

Reviewer's Responses to Questions

**Comments to the Author**

1. Does the manuscript provide a valid rationale for the proposed study, with clearly identified and justified research questions?

Reviewer #1: Partly

Reviewer #2: Yes

2. Is the protocol technically sound and planned in a manner that will lead to a meaningful outcome and allow testing the stated hypotheses?

Reviewer #1: Partly

Reviewer #2: Partly

3. Is the methodology feasible and described in sufficient detail to allow the work to be replicable?

Reviewer #1: Yes

Reviewer #2: Yes

4. Have the authors described where all data underlying the findings will be made available when the study is complete?

Reviewer #1: Yes

Reviewer #2: No

5. Is the manuscript presented in an intelligible fashion and written in standard English?

Reviewer #1: Yes

Reviewer #2: Yes

6. Review Comments to the Author

You may also provide optional suggestions and comments to authors that they might find helpful in planning their study.

Reviewer #1: Thank you for allowing me to review this manuscript related to a protocol for a scoping review. The following may assist in improving the manuscript further:

Background: Although the concept of people who are socially disadvantaged is well-explained in the inclusion/exclusion criteria in the methodology, perhaps a definition early in the background could be provided as the concept of people who are socially disadvantaged could have different meanings in different contexts.

Objectives: the justification for conducting the scoping review is clear. However, this justification could be strengthened by indicating whether any reviews have been conducted on the topic or related topics and where gaps are.

Methods: although the inclusion and exclusion are well-described in terms of the concept and language, perhaps also to include what type of articles/studies/literature will be searched for.

Limitations: limitations were clearly indicated. Perhaps also to indicate what measures/strategies will be implemented to enhance rigour of the scoping review.

Documentation/reporting: the how of reporting of the review was outlined, however perhaps the reporting tools (e.g. PRISMA flowchart, tables, graphs) could be included.

Reviewer #2: COMMENTS TO THE AUTHORS

This scoping review seeks to identify the services, models of care and interventions developed to improve access to cancer treatment for socially disadvantaged people. The study is relevant and can contribute to overcoming the barriers that socially disadvantaged people face.

However, the protocol requires minor modification/revision (specified below) before consideration for publication.

MAJOR ISSUES

Line 146: Indicate whether studies will be included or excluded based on their design and provide a rationale. For instance, will the authors include both experimental and non-experimental interventions that improve access to cancer treatment?

Line 155: Study selection: Describe how duplicate records/studies will be identified and managed.

Line 166: Table 1 (Inclusion and Exclusion Criteria) suggests that the review will focus on adults aged 18 years and older who are socially disadvantaged. This focus must reflect in the background, abstract and possibly the title of the review. Socially disadvantaged people may include children, adolescents or youth etc.

Line 166: Focus of publication: The authors indicated that “Focus is on describing a service, intervention or models of care developed and implemented with the explicit aim of improving access to, receipt of, or adherence to cancer treatment”. The concept “adherence” is introduced here for the first time in the protocol. Adherence to cancer treatment is absent in the review question, purpose or search strategy. Countless services, models of care and interventions (experimental/non-experimental) that “improve adherence to cancer treatment” exist and will considerably expand the scope and resources required of the review.

Consider deleting the “adherence to cancer treatment” component. Otherwise, comprehensively integrate the “adherence to cancer treatment” component into the scoping review.

MINOR ISSUES

Line 36: Review the sentence “A comprehensive search strategy will be developed by an academic librarian will be developed”.

Line 42: Consider removing the open science registration link from the abstract.

Line 55: Review sentence starting with “Evidence also….”

Line 146: Identifying relevant studies: Indicate what action(s) will be taken if the full text of a record/study (other than grey literature and dissertations) is not found.

Table 166: Exclusion criteria: Provide a rationale for studies published in languages other than English language.

Line 166: Consider merging the two population rows.

7. PLOS authors have the option to publish the peer review history of their article (what does this mean?). If published, this will include your full peer review and any attached files.

Reviewer #1: No

Reviewer #2: No

---

## [Author Response · Author response to Decision Letter 0]

18 Aug 2023

We thank the reviewers for their thoughtful and constructive comments. We have included a point by point response to feedback received in a table uploaded as 'Response to Reviews'.

---

## [Decision Letter · Decision Letter 1]

23 Oct 2023

PONE-D-23-11579R1Services, models of care, and interventions to improve access to cancer treatment for adults who are socially disadvantaged: A scoping review protocolPLOS ONE

Dear Dr. Horrill,

Thank you for submitting your manuscript to PLOS ONE. After careful consideration, we feel that it has merit but does not fully meet PLOS ONE’s publication criteria as it currently stands. Therefore, we invite you to submit a revised version of the manuscript that addresses the points raised during the review process. The manuscript requires minor changes as suggested by the reviewer. Please submit the revised manuscript.

We look forward to receiving your revised manuscript.

Kind regards,

AKM Alamgir, PhD

Academic Editor

PLOS ONE

Journal Requirements:

Reviewers' comments:

Reviewer's Responses to Questions

**Comments to the Author**

1. Does the manuscript provide a valid rationale for the proposed study, with clearly identified and justified research questions?

Reviewer #1: Yes

Reviewer #3: Yes

2. Is the protocol technically sound and planned in a manner that will lead to a meaningful outcome and allow testing the stated hypotheses?

Reviewer #1: Yes

Reviewer #3: Yes

3. Is the methodology feasible and described in sufficient detail to allow the work to be replicable?

Reviewer #1: Yes

Reviewer #3: Yes

4. Have the authors described where all data underlying the findings will be made available when the study is complete?

Reviewer #1: Yes

Reviewer #3: Yes

5. Is the manuscript presented in an intelligible fashion and written in standard English?

Reviewer #1: Yes

Reviewer #3: Yes

6. Review Comments to the Author

You may also provide optional suggestions and comments to authors that they might find helpful in planning their study.

Reviewer #1: Thank you for addressing the feedback. I would hereby recommend to accept the manuscript for publication.

Reviewer #3: The authors have identified a very important gap with respect to the need to examine the existing literature on services, models of care and interventions to improve access to cancer treatment for people who are socially disadvantaged through conducting a scoping a review. The protocol is well articulated and fits the aims of the review accordingly. I would just suggest to consider anchoring the definition of the population "socially disadvantaged" which is well done in the methods and population definition in the body of the paper to be done early on in the manuscript in the abstract as well as introduction so that the reader has a very clear understanding of how your population is defined. Another suggestion would be to provide a bit more context to geography and region - we see that the literature will be searched for english language having said that is there an assumption that the context is North America or international in terms of how we define populations globally with respect to socially disadvantaged and cancer screening as the multiple components of how you define social disadvantage cannot be separated from national/geographical and political contexts that may vary globally. Minor detail - the appendices are referred differently in the body - ie. appendix A in the body is actually Appendix B ~

This is a timely scoping review protocol and addresses a gap that is of importance to advancing our understanding of the needs of people experiencing marginalization.

7. PLOS authors have the option to publish the peer review history of their article (what does this mean?). If published, this will include your full peer review and any attached files.

Reviewer #1: No

Reviewer #3: **Yes: **Rosanra Yoon

---

## [Author Response · Author response to Decision Letter 1]

25 Oct 2023

We thank the reviewers for their thoughtful suggestions. We have made minor revisions, and have outlined our response to the reviewer suggestions in the document uploaded "Response to Reviews".

---

## [Editor Report · Decision Letter 2]

18 Dec 2023

Services, models of care, and interventions to improve access to cancer treatment for adults who are socially disadvantaged: A scoping review protocol

PONE-D-23-11579R2

Dear Dr. Tara Horrill,

We’re pleased to inform you that your manuscript has been judged scientifically suitable for publication and is formally accepted for publication.

Kind regards,

AKM Alamgir, PhD

Academic Editor

PLOS ONE
---

## [Editor Report · Acceptance letter]

16 Feb 2024

PONE-D-23-11579R2 

PLOS ONE

Dear Dr. Horrill, 

I'm pleased to inform you that your manuscript has been deemed suitable for publication in PLOS ONE. Congratulations! Your manuscript is now being handed over to our production team.

Kind regards, 

on behalf of

Dr AKM Alamgir 

Academic Editor

PLOS ONE